# Effect of Gd^3+^ Substitution on Thermoelectric Power Factor of Paramagnetic Co^2+^-Doped Calcium Molybdato-Tungstates

**DOI:** 10.3390/ma14133692

**Published:** 2021-07-01

**Authors:** Bogdan Sawicki, Marta Karolewicz, Elżbieta Tomaszewicz, Monika Oboz, Tadeusz Groń, Zenon Kukuła, Sebastian Pawlus, Andrzej Nowok, Henryk Duda

**Affiliations:** 1Institute of Physics, University of Silesia in Katowice, 40-007 Katowice, Poland; bogdan.sawicki@us.edu.pl (B.S.); monika.oboz@us.edu.pl (M.O.); zenon.kukula@us.edu.pl (Z.K.); sebastian.pawlus@us.edu.pl (S.P.); andrzej.nowok@smcebi.edu.pl (A.N.); henryk.duda@us.edu.pl (H.D.); 2Faculty of Chemical Technology and Engineering, West Pomeranian University of Technology in Szczecin, 70-310 Szczecin, Poland; marta_pawlikowska@op.pl (M.K.); tomela@zut.edu.pl (E.T.)

**Keywords:** sintering, scheelites, magnetic properties, electrical properties, dielectric spectroscopy

## Abstract

A series of Co^2+^-doped and Gd^3+^-co-doped calcium molybdato-tungstates, i.e., Ca_1−3x−y_Co_y_
_x_Gd_2x_(MoO_4_)_1−3x_(WO_4_)_3x_ (CCGMWO), where 0 < *x* ≤ 0.2, *y* = 0.02 and represents vacancy, were successfully synthesized by high-temperature solid-state reaction method. XRD studies and diffuse reflectance UV–vis spectral analysis confirmed the formation of single, tetragonal scheelite-type phases with space group *I*4_1_/*a* and a direct optical band gap above 3.5 eV. Magnetic and electrical measurements showed insulating behavior with *n*-type residual electrical conductivity, an almost perfect paramagnetic state with weak short-range ferromagnetic interactions, as well as an increase of spin contribution to the magnetic moment and an increase in the power factor with increasing gadolinium ions in the sample. Broadband dielectric spectroscopy measurements and dielectric analysis in the frequency representation showed a relatively high value of dielectric permittivity at low frequencies, characteristic of a space charge polarization and small values of both permittivity and loss tangent at higher frequencies.

## 1. Introduction

Scheelite-type molybdates and tungstates are attractive materials due their large efficient scintillation yield, X-ray absorption coefficient, and high thermal and chemical stability. A number of these compounds have been developed in the field of scintillation detectors (e.g., CaMoO_4_ or PbWO_4_), phosphors, and laser materials [1,2,3,4,5].

Recently, studies of manganese doped calcium molybdato-tungstates with the chemical formula of Ca_1−x_Mn_x_(MoO_4_)_0.50_(WO_4_)_0.50_ (*x* = 0.01, 0.03, 0.05, 0.10, 0.125, and 0.15) showed a weak *p*-type electrical conductivity and the paramagnetic state with a change in the short-range interactions from ferromagnetic to antiferromagnetic as well as an increase in the orbital contribution to the magnetic moment with increasing Mn^2+^ content [6]. Next, studies of Gd^3+^-doped lead molybdato-tungstates with the chemical formula of Pb_1−3x_
_x_Gd_2x_(MoO_4_)_1−3x_(WO_4_)_3x_ (*x* = 0.0455, 0.0839, 0.1154, 0.1430, 0.1667, and 0.1774; ↓ represents vacancies) showed a paramagnetic state with characteristic superparamagnetic-like behavior. Moreover, they revealed faster and slower relaxation processes in the loss spectra with various time scales for the gadolinium-poorer samples and no signs of any relaxation processes for the gadolinium-richer ones [7]. The above studies prove that 3*d* transition metal ions or 4*f* rare earth ones significantly affect the electrical and magnetic properties of molybdato-tungstates.

Recently, studies of manganese doped calcium molybdato-tungstates with the chemical formula of Ca_1−x_Mn_x_(MoO_4_)_0.50_(WO_4_)_0.50_ (*x* = 0.01, 0.03, 0.05, 0.10, 0.125, and 0.15) showed a weak *p*-type electrical conductivity and the paramagnetic state with a change in the short-range interactions from ferromagnetic to antiferromagnetic as well as an increase in the orbital contribution to the magnetic moment with increasing Mn^2+^ content [6]. Next, studies of Gd^3+^-doped lead molybdato-tungstates with the chemical formula of Pb_1−3x_
_x_Gd_2x_(MoO_4_)_1−3x_(WO_4_)_3x_ (*x* = 0.0455, 0.0839, 0.1154, 0.1430, 0.1667, and 0.1774; ↓ represents vacancies) showed a paramagnetic state with characteristic superparamagnetic-like behavior. Moreover, they revealed faster and slower relaxation processes in the loss spectra with various time scales for the gadolinium-poorer samples and no signs of any relaxation processes for the gadolinium-richer ones [7]. The above studies prove that 3*d* transition metal ions or 4*f* rare earth ones significantly affect the electrical and magnetic properties of molybdato-tungstates.

The motivation for this work is to search for materials useful in electronics and technology. For this reason, the selection of appropriate elements is essential. Rare earth ions have strong electron screening on 4*f* shells and are paramagnetic ions. Gadolinium ion (Gd^3+^, 4f^5^) has a high magnetic moment due to its isotropic electronic ground state of _7/2_S and excellent magnetic resonance imaging effect used as a common contrast agent [8]. In turn, *d*-electron metal ions with unfilled 3*d*-shells, such as Co^2+^, have a bigger capacity for charge accumulation [9,10]. The concentration of *x* and *y* ions is limited by the phase nature of the compound. 

That is why we propose electrical and magnetic investigations of new molybdato-tungstates containing both 3*d* and 4*f* ions. New calcium molybdato-tungstates doped with cobalt and gadolinium ions with the chemical formula of Ca_1−3x−y_Co_y_
_x_Gd_2x_(MoO_4_)_1−3x_(WO_4_)_3x_ (CCGMWO), where *x* = 0.0050, 0.0098, 0.0238, 0.0455, 0.0839, 0.1430, 0.1667, and 0.2000; *y* = 0.02 and denotes vacancies, were successfully synthesized via high-temperature annealing of ternary CoMoO_4_/Gd_2_(WO_4_)_3_/CaMoO_4_ mixtures with various content of initial reactants. The concentration range of Gd^3+^ ions in materials under study was selected to cover the homogeneity range of new solid solution, i.e., the range when 0 < *x* ≤ 0.2000 and 0 < *y* ≤ 0.02. The doped materials adopt the tetragonal scheelite-type structure with space group *I*4*_1_/a*. 

## 2. Experimental Details

### 2.1. Synthesis of CCGMWO Solid Solution 

Microcrystalline samples of new solid solution were obtained by two-step synthesis. In both steps, high-temperature sintering of appropriate materials was applied. The following reactants were used in the first step of synthesis: CaCO_3_, CoCO_3_, MoO_3_, Gd_2_O_3_, and WO_3_ (all raw materials of high-purity grade min. 99.95%, Alfa Aesar and without thermal pre-treatment). Calcium molybdate (CaMoO_4_), cobalt molybdate (CoMoO_4_), and gadolinium tungstate (Gd_2_(WO_4_)_3_) were obtained according to the procedure used in our previous studies [11,12]. In the next step of the synthesis, we prepared eight ternary mixtures containing CoMoO_4_, Gd_2_(WO_4_)_3_, and CaMoO_4_. The concentration of cobalt molybdate was constant in each initial CoMoO_4_/Gd_2_(WO_4_)_s_/CaMoO_4_ mixture, taking the value of 3.00 mol%. The content of Gd_2_(WO_4_)_3_ was variable and ranged from 0.50 to 33.33 mol%. The initial mixtures, not compacted to pellets, were heated at temperatures in the range of 1173–1473 K in air, in several 12 h annealing stages. After each sintering period, obtained samples were cooled down to ambient temperature, weighed, ground in a porcelain mortar, and followed by examination for their composition by XRD method. We observed a slight mass loss for each obtained material. This loss did not exceed the value of 0.25%. This observation shows that a synthesis of doped samples runs practically without their mass change and a process between initial reactants occurred according to the general equation: (1−3*x*−*y*)·CaMoO_4_ + *x*·Gd_2_(WO_4_)_3_ + *y*·CoMoO_4_ = Ca_(1−3*x*−*y*)_Co*_y_*
*_x_*Gd_2*x*_(MoO_4_)_(1−3*x*)_(WO_4_)_3_, where homogeneity range of this solid solution is 0 < *x* ≤ 0.2 and 0 < *y* ≤ 0.02. When initial CoMoO_4_ content was *y* = 0.02, the chemical reaction of initial reactants can be described as follows: (0.98−3*x*)·CaMoO_4_ + *x*·Gd_2_(WO_4_)_3_ + 0.02·CoMoO_4_ = Ca_(0.98−3*x*)_Co_0.02_
*_x_*Gd_2*x*_(MoO_4_)_(1−3*x*)_(WO_4_)_3*x*_.

### 2.2. Characterization of Methods

X-ray diffraction (XRD) patterns of obtained powder samples were recorded within the 2*Θ* range of 10–100° on an EMPYREAN II diffractometer (PANalytical, Malvern, UK) with CuKα_1,2_ radiation (λ_aver_ = 1.5418 Å, the scanning step 0.013°). High-Score Plus 4.0 software was used to analyze collected XRD patterns. Lattice parameters were calculated using DICVOL04 software [13]. The density of the samples under study was measured using a ultrapycnometer, model Ultrapyc 1200 e (Quantachrome Instruments, Boynton Beach, USA). A pycnometric gas nitrogen (purity 99.99%) was applied. 

Particle size and size distribution were measured using a laser particle size analyzer Mastersizer 3000 (Malvern Instruments Ltd., Malvern, UK), dispersing the particles in water by an ultrasonic bath. For each sample, 10 readings were accounted to allow the statistical analysis. The collected data were processed with the equipment’s software, which calculates particle size and size distribution per percentage of volume. The size distribution was estimated using D_10_, D_50_, and D_90_ which indicate the volume fraction of the sample under a certain size (µm). 

UV–vis reflectance spectra were recorded within the spectral range of 200–1000 nm using a JASCO-V670 spectrophotometer (JASCO Europe S.R.L., Cremella, Italy) equipped with an integrating sphere.

The static dc magnetic susceptibility was measured in the temperature range of 2–300 K. Magnetization isotherms were measured at 2, 10, 20, 40, 60, and 300 K using a Quantum Design MPMS-XL-7AC SQUID magnetometer (Quantum Design, San Diego, CA, USA) in applied external fields up to 70 kOe. The electrical conductivity *σ*(*T*) was measured by the DC method using KEITHLEY 6517B Electrometer/High Resistance Meter (Keithley Instruments, LLC, Solon, OH, USA) in the temperature range of 300–400 K. The thermoelectric power *S*(*T*) was measured in the temperature range of 300–600 K using Seebeck Effect Measurement System (MMR Technologies, Inc., San Jose, CA, USA). Dielectric properties of the solid solution under study were measured using a Broadband Dielectric Spectrometer (Novocontrol Technologies GmbH & Co. KG, Montabaur, Germany) equipped with an Alpha Impedance Analyzer with Active Sample Cell and Quatro Cryosystem temperature control. Investigations were performed when a temperature decreased from 373 K to 173 K with the *T*-step of 5 K. The calculations of the effective moment, Landé factor using the Brillouin procedure and the imaginary part of the complex permittivity using the Kramers–Kronig transform are described in detail in [9,14,15,16,17,18]. The electrical and thermal contacts were made using a silver lacquer mixture (Degussa Leitsilber 2000, is Degussa Gold und Silber, Munich, Germany).

## 3. Results and Discussion

### 3.1. X-ray Diffraction Analysis and Particle Size Distribution

The powder XRD patterns of pure CaMoO_4_ and CCGMWO solid solution with different content of Gd^3+^ ions, i.e., when *x* = 0.0050, 0.0098, 0.0283, 0.0455, 0.0839, 0.1430, 0.1667, 0.2000, and the constant concentration of Co^2+^, i.e., *y* = 0.02 are shown in Figure 1a,b. XRD analysis shows that the powder diffraction patterns of new Co^2+^-doped and Gd^3+^-co-doped calcium molybadato-tungstates consisted of diffraction lines. These lines can be attributed to scheelite-type framework. No impurity phases, i.e., initial reactants, metal oxides, and other gadolinium tungstates or molybdates were observed with increasing gadolinium ions concentration only up to *x* = 0.2 (33.33 mol% of Gd_2_(WO_4_)_3_ in initial CoMoO_4_/Gd_2_(WO_4_)_3_/CaMoO_4_ mixtures when *y* is constant and equals 0.02, i.e., 3.00 mol% of CoMoO_4_). The observed diffraction lines attributed to the scheelite-type structure kept their position or shifted slightly towards the lower 2Θ angle with increasing Gd^3+^ content. All registered peaks were successfully indexed to the pure tetragonal scheelite-type structure with space group *I*4_1_/*a* (No. 88, CaMoO_4_ − JCPDs No. 04-013-6763). This fact confirmed the formation of a new solid solution of CoMoO_4_ and Gd_2_(WO_4_)_3_ in CaMoO_4_ matrix. The diffraction patterns of samples comprising initially above 33.33 mol% Gd_2_(WO_4_)_3_ in initial CoMoO_4_/Gd_2_(WO_4_)_3_/CaMoO_4_ mixtures (not shown in our manuscript) revealed simultaneously presence of peaks attributed to the saturated solid solution (when *x* = 0.2 and *y* = 0.02) as well as the diffraction lines characteristic Gd_2_(WO_4_)_3_. It means that the solubility limit of gadolinium tungstate in CaMoO_4_ crystal lattice when the initial concentration of CoMoO_4_ was 3.00 mol% is not higher than 33.33 mol%.

Table 1 shows lattice constants calculated for CaMoO_4_ and samples of new solid solution. Both unit cell parameters of doped materials change non-linearly with increasing Gd^3+^ concentration. The observed changes of *a* and *c* lattice constants are not identical (Figure 2). Initially, the *a* parameter increases, and then its value decreases with increasing content of Gd^3+^ ions. The *c* parameter shows an opposite dependence. The unit cell volume shows an increasing tendency with the increasing amount of Gd^3+^ ions in the crystal lattice of solid solution (Table 1). This is not a typical situation because bigger Ca^2+^ ions (ionic radius − 1.12 Å for CN = 8) in CaMoO_4_ matrix were simultaneously substituted by much smaller Co^2+^ (0.90 Å for CN = 8) and Gd^3+^ (1.053 Å for CN = 8) ones [19]. We have already observed a similar phenomenon in other RE^3+^-doped and vacancied molybdates, i.e., Cd_1−3x_
_x_Dy_2x_MoO_4_ [20]. In scheelite-type molybdates and tungstates with the chemical formula of AXO_4_ (A—divalent metal; X = Mo or W), Mo^6+^ and W^6+^ ions are tetrahedrally coordinated by O^2−^ ones and their ionic radii are 0.41 and 0.42 Å, respectively [19]. Thus, the substitution of Mo^6+^ ions by W^6+^ ones observed while forming new solid solution did not cause any significant changes in both lattice parameters. We also calculated the lattice parameter ratio *c*/*a* (Table 1). This parameter clearly changes nonlinearly with increasing *x*, and it reaches the minimum value (*c*/*a* = 2.1798) for *x* = 0.1430. It means that in the whole homogeneity range of CCGMWO solid solution we have observed the deformation of the tetragonal scheelite-type cell of each sample in comparison to the pure matrix’s tetragonal cell. Figure 3 shows the visualization of scheelite-type structure of the CCGMWO solid solution. In this type of structure, Ca^2+^/Co^2+^/Gd^3+^/ and Mo/W^6+^ ions occupy the *S*_4_ sites, but the oxygen ions are at *C*_1_ sites. Each Mo/W^6+^ ion is surrounded by four oxygen ions (CN = 4), but each Ca^2+^/Co^2+^/Gd^3+^ or vacancy site is surrounded by eight oxygen ions (CN = 8). The substituted ions and vacancies are randomly distributed in CaMoO_4_ matrix for low Gd^3+^ concentration, i.e., when 0 < *x* ≤ 0.1430 and statistically distributed for x ≥ 0.1667. The experimental density values determined for samples under study are given in Table 1. The density of CCGMWO materials almost linearly increases with increasing Gd^3+^ content.

The particle size distribution of doped powder materials in volume is shown in Figure 4. The obtained curves exhibit a typical bimodal distribution with one very weak peak between 4 and 20 µm. A bimodal distribution of particle size is observed for all Gd^3+^ concentrations, and this distribution is typical for annealing processes. The maximum values of particle size, D_90_, increases with increasing Gd^3+^ concentration in new Co^2+^-doped and Gd^3+^-co-doped calcium molybdato-tungstates from ~90 µm (*x* = 0.0050, *y* = 0.02) to ~120 µm for the saturated solid solution (*x* = 0.2, *y* = 0.02).

### 3.2. UV –Vis Spectra and Their Analysis 

Divalent metal molybdates and tungstates with a scheelite-type structure such as CaMoO_4_ or CaWO_4_ have a typical optical absorption process characterized by a direct electronic transition occurring in maximum energy states near or in the valence band to minimum energy states located below or in the conduction band [21,22]. Due to the lack of impurity levels between the valence band and conduction one, molybdates and tungstates have a high band gap (*E*_g_). This value is associated with a degree of order and disorder structural these materials. Optical properties of Co^2+^-doped and Gd^3+^-co-doped calcium molybdato-tungstates were studied by UV–vis diffused reflectance spectroscopy. Using the Kubelka–Munk method, the recorded UV–vis reflectance spectra were converted into absorption ones [23]. Figure 5a displays the optical absorption spectra of CCGMWO samples collected within the spectral range of 200–1000 nm. The Tauc’s model was used to calculate the values of direct band gap (*E*_g_) using the following Equation (1) [24,25,26]:(*αhν*)^2^ = A(*hν* − *E*_g_)(1)
where *α* is absorption coefficient, A is a constant that depends on the transition nature, and *hν* is photon energy (in eV). The optical band gap was determined by plotting a graph between (*αhν*)^2^ versus *hν* and extrapolation of straight to (*αhν*)^2^ = 0 (Figure 5b). Its values for pure matrix and all doped samples are listed in Table 1. It was observed that the optical band gap of CCGMWO decreased when Gd^3+^ concentration increased from *x* = 0.0050 up to *x* = 0.1430. For the latter Gd^3+^ content, we observed the lowest value of *E*_g_, i.e., 3.51 eV. The systematic decrease in an optical band with Gd^3+^ doping is attributed to structural disorder and on-site fluctuations, which arise due to the substitution of Co^2+^ and Gd^3+^ ions for Ca^2+^ ones, W^6+^ for Mo^6+^, as well as a creation of vacancies. Additionally, there is a quite significant electronegativity difference between Ca (1.00), Gd (1.20), and Co (1.88) as well as between Mo (2.16) and W (2.36). The differences in electronegativity values shift the valence band towards the conduction band and lead to a decrease in a band gap with doping. On the other hand, the samples with more significant Gd^3+^ ions concentration, i.e., when *x* > 0.1430 direct band gap systematically increase. This means that a consistently growing number of defects in the crystal lattice of CCGMWO solid solution are ordered. This structure is rearranged so as to be stable and regular.

The incorporation of doping ions into the structure of some materials often reveals a formation of band tailing in the band gap due to a formation of localized states. The band tail energy (Urbach energy, *E*_U_) characterizes the width of located states, and the empirical Urbach’s relation express it [27]:*α* = *α*_0_ exp (*h**ν*/*E*_U_)(2)
where *α* is an experimentally determined absorption coefficient and *α*_0_ is a constant. Urbach energy can be obtained from a slope of the straight line of plotting ln(*α*) against photon energy (*hν*). Figure 5b (inset)shows the variation of ln(*α*) versus photon energy of the sample when *x* = 0.0455. *E*_U_ values for samples under study were calculated from reciprocal of the straight line slopes, as shown in Figure 5b, and they are collected in Table 1. The presence of a band tail for CCGMWO samples indicates that the structural inhomogeneous and disorder are significant. Similarly, Figure 6 displaying *E*_g_ and *E*_U_ variations as a function of Gd^3+^ concentration shows that both *E*_g_ and *E*_U_ values correlate very well. It is very clearly seen from Figure 6 that the optical band gap values are opposite to the degree of disorder in a structure. As a result, both a decrease in the optical gap and a broadening of the Urbach tail occurred. The lowest value of *E*_g_ (3.51 eV) we observed for the highest Urbach energy, i.e., when *E*_U_ = 0.578 eV.

### 3.3. Magnetic Properties

Magnetic susceptibility (Figure 7a,b) and its inverse (Figure 7c) measurements of CCGMWO materials showed an almost perfect paramagnetic state with weak short-range ferromagnetic interactions demonstrated by low Curie–Weiss paramagnetic temperature values θ ~ 1 K (Table 2), as well as no splitting between ZFC and FC magnetic susceptibility (Figure 7a,b) for any phase. These facts are suggesting no spin frustration and no long-range magnetic interactions in the studied temperature range. The Brillouin fitting procedure showed an increase of spin contribution to the magnetic moment with increasing content of gadolinium ions in the sample because the Landé factor, *g*, goes to 2 (Figure 7, Table 2). In the reduced coordinate (H/T), magnetic isotherms fall exactly on a universal Brillouin curve for all samples under study (Figure 8). Such behavior is characteristic of superparamagnetic particles. Similar behavior was found for the following single crystals: CdMoO_4_:Gd^3+^ [28], CdMoO_4__WO_4_:Yb^3+^ [29], as well as ZnGd_4_W_3_O_16_ [9] and Gd_2_W_2_O_9_ [30] microcrystalline materials. Comparable values of the effective magnetic moment, µ_eff_, and the effective number of Bohr magnetons, *p*_eff_, showed that the cobalt ions contribute a small orbital contribution to the magnetic moment (Table 2). Therefore, poorer gadolinium samples have a stronger spin-orbit coupling effect than the richer ones. A similarly strong spin contribution to the magnetic moment was observed, for example, in Gd-doped superparamagnetic lead molybdate-tungstate microcrystals [7] and nanocrystals [31] as well as CdMoO_4_:Gd^3+^ single crystals [28].

### 3.4. Electrical Properties

The results of the electrical conductivity (Figure 9) and thermoelectric power (Figure 10) measurements of CCGMWO microcrystals showed insulating behavior with small values of the *n*-type residual conductivity of σ ~ 10^−8^ S/m. No thermal conductivity activation of the current carriers for poorer gadolinium samples was observed. This thermal activation is only visible above room temperature for samples richer in gadolinium. In general, the residual electrical conduction of the *n*-type CCGMWO materials under study seems to be connected with the anionic vacancies. It’s known that in a state of thermal equilibrium, structural defects (*n*) are always present in the lattice, even in the crystal which is ideal in other respects. A necessary condition for free energy minimalization gives *n* ≅ *N*exp(-*E*v/*k*B*T*) for *n* << *N*, where *N* is the number of atoms in the crystal and *E*v is the energy required to transfer the atom from the bulk of the crystal on its surface [32]. Similar behavior for the following Gd^3+^-doped lead molybdato-tungstates [28], RE^3+^-doped tungstates or molybdates: R_2_WO_6_ (R = Nd, Sm, Eu, Gd, Dy, Ho) [33], CdRE_2_W_2_O_10_ (RE = Y, Pr, Nd, Sm, Gd–Er) [10,34], RE_2_W_2_O_9_ (RE = Pr, Sm–Gd) [35], AgY_1–*x*_Gd*_x_*(WO_4_)_2_ (*x* = 0.005, 0.01, 0.025, 0.04, 0.10, 0.20, 1.00) [36], and Cd_1–3*x*_Gd_2*x*_
*_x_*MoO_4_ (0.0005 ≤ *x* ≤ 0.0455) [36] were observed.

Figure 11 shows an interesting dependence of the power factor S^2^σ on temperature *T*. The power factor has a very small value of a few fW/(cmK^2^). However, its value will significantly increase with the increase in temperature for samples richer in gadolinium ions. Usually, a low value of the power factor suggests rather the hopping transport, magnon scattering, and lower covalence of the studied materials compared to classical thermoelectric semiconductors [37]. Materials with a large power factor value (S^2^σ) are usually heavily doped semiconductors, such as Bi_2_Te_3_ [37]. The above studies show that even in ion-bonded materials, thermoelectric efficiency can be improved by doping them with rare-earth ions. The observed strong dependence of the power factor on the content of gadolinium ions may result from the size effect, as the Gd^3+^ ions have a large ion radius [19].

Broadband dielectric spectroscopy measurements of the solid solution under study showed both small values of the real part (ε′) and imaginary part (ε″) of complex dielectric permittivity strongly decreasing with frequency and temperature (Figure 12a–d). Dielectric analysis in the frequency representation showed that no dipole relaxation processes were observed (as in Maxwell-Wagner [38] or Jonscher [39]) on the real and imaginary spectra of representative samples with the content *x* = 0.0050 (a), 0.0455 (b), 0.0839 (c), and 0.2000 (d) (Figure 12). Only in the case of the sample with the lowest gadolinium content (*x* = 0.0050) the collected spectra indicated the presence of the relaxation process (characteristic “bulge” on the loss curve, indicated by an arrow, shifting to lower frequencies with cooling). However, calculated data from the Kramers–Kronig transform [18] of ε′ did not reveal a well-defined relaxation peak (Figure 13). The calculated loss tangent values indicated that all samples except *x* = 0.0050 have tan*δ* ~0.01, in which the relaxation process is inhibited with increasing gadolinium content in the sample (Figure 14). Both small ε′ and tan*δ* values can be explained by the fact that Gd^3+^ ions have the 4*f*-shell screened. This hinders the formation of electric dipoles and their relaxation. Low energy loss was also observed in Ca_1−x_Mn_x_MoO_4_ nanomaterials (*x* = 0.0, 0.01, 0.05, 0.10, 0.15) [40] and in RE_2_W_2_O_9_ ceramics (RE = Pr, Sm-Gd) [35]. It is worth noting that significantly higher electric permittivity values at low frequencies are characteristic of the polarization of the space charge induced in the sample.

## 4. Conclusions

In summary, the samples of CCGMWO solid solution were characterized by optical, magnetic, electrical, and dielectric spectroscopy measurements. They have shown perfect paramagnetic state and superparamagnetic behavior as well as insulating properties with the *n*-type residual conductivity. Broadband dielectric spectroscopy measurements and dielectric analysis in the frequency representation showed the space charge polarization and small values of both the real and imaginary part of permittivity and a lack of dipole relaxation. Such small values of the loss tangent of the materials under study give them potential applications for the producing of lossless capacitors. A significant achievement of these studies is the observation of an increase in the thermoelectric power factor with an increase in the gadolinium ions content in the sample.

## Figures and Tables

**Figure 1 materials-14-03692-f001:**
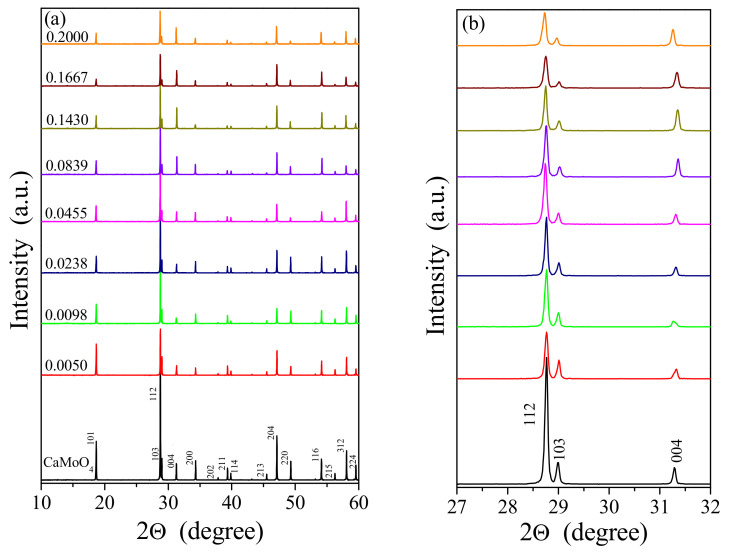
(**a**) Powder XRD patterns of CaMoO_4_ and CCGMWO ceramics within the 2Θ range of 10–60°; (**b**) 112/103/004 diffraction lines in the range of 2Θ from 27° to 32°.

**Figure 2 materials-14-03692-f002:**
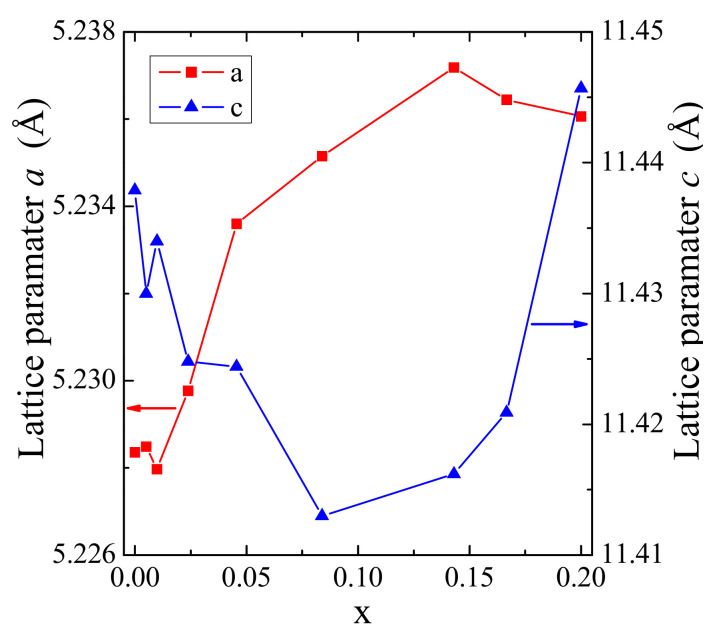
Variation of *a* and *c* unit cell constants as a function of Gd^3+^ doping in CCGMWO materials.

**Figure 3 materials-14-03692-f003:**
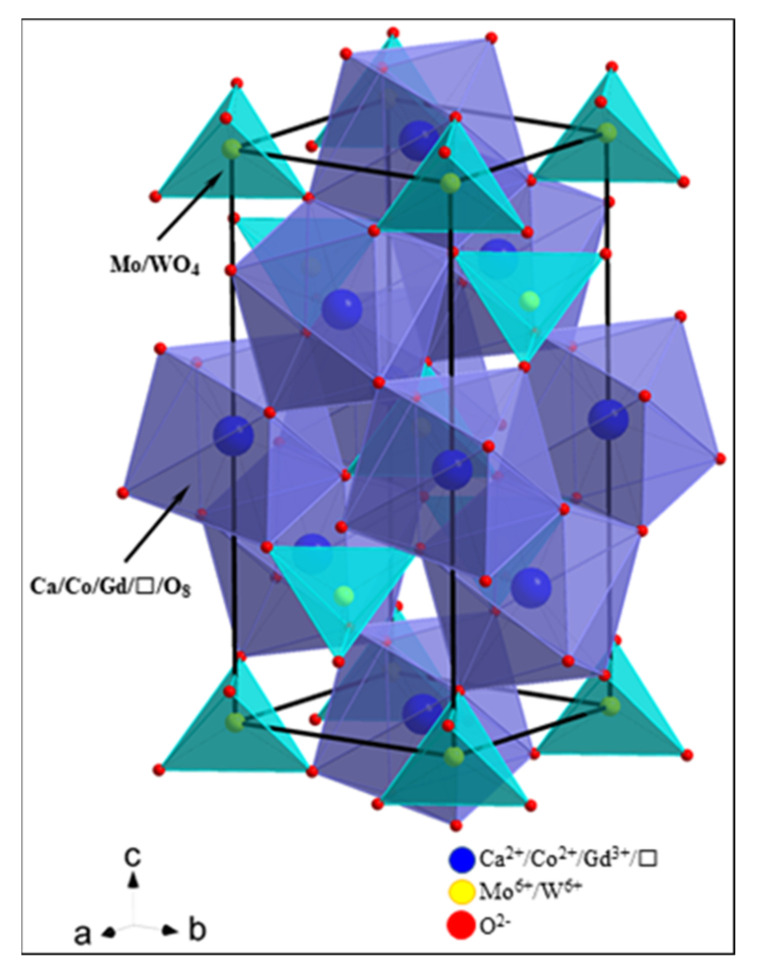
Visualization of the scheelite-type structure of the CCGMWO solid solution. Legend: navy balls—Ca^2+^/Co^2+^/Gd^3+^/ ; yellow balls—Mo(W)^6+^; red balls—O^2−^; and a, b, and c—crystallographic axes.

**Figure 4 materials-14-03692-f004:**
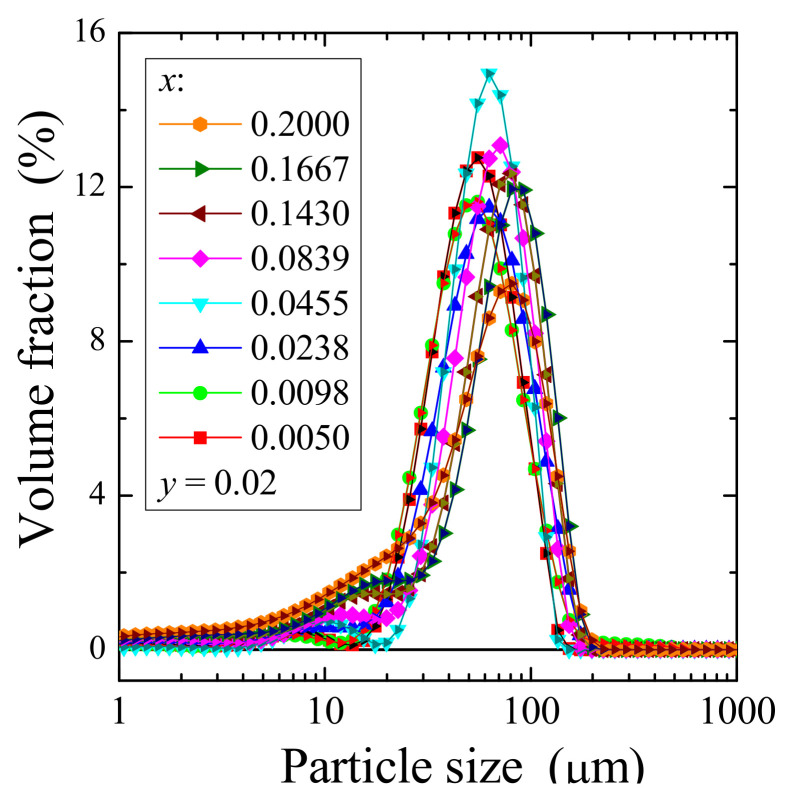
Particle size distribution of CCGMWO samples.

**Figure 5 materials-14-03692-f005:**
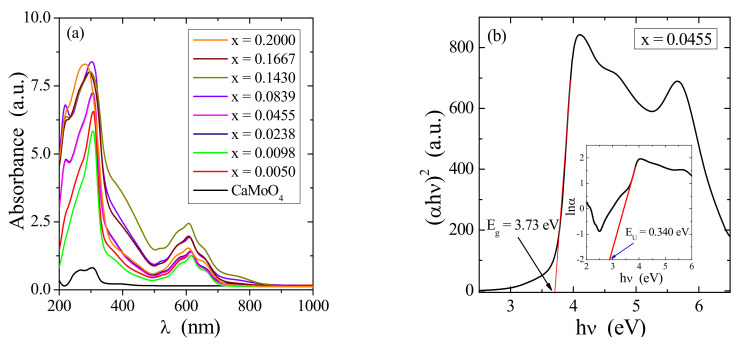
(**a**) UV–vis absorption spectra of CaMoO_4_ and CCGMWO samples; (**b**) Plot of (αhν)^2^ vs. hν for CCGMWO (x = 0.0455, y = 0.02) and determined band gap energy; (insert)—plot of ln(α) vs. hν for this sample and determined Urbach energy.

**Figure 6 materials-14-03692-f006:**
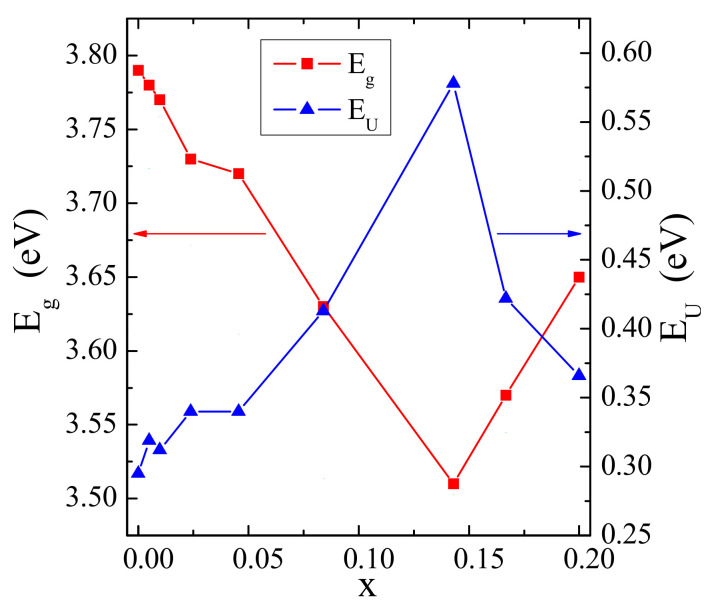
Variation of direct band gap (E_g_) and Urbach energy (E_U_) with Gd^3+^ doping in CCGMWO samples.

**Figure 7 materials-14-03692-f007:**
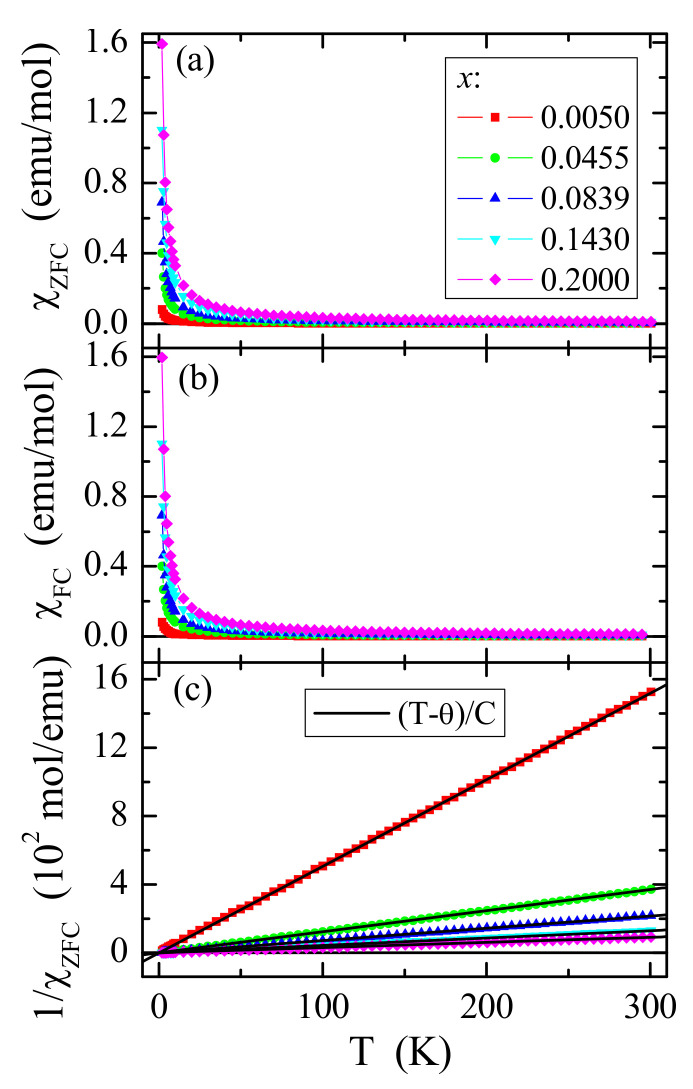
χ_ZFC_ (**a**), χ_FC_ (**b**) dc magnetic susceptibility and its inverse 1/χ_ZFC_ (**c**) vs. temperature *T* of CCGMWO solid solution for *x* = 0.0050, 0.0455, 0.0839, 0.1430, 0.2000, and *y* = 0.02 recorded at *H* = 1 kOe. The solid (black) line in (**b**), (T-θ)/C, indicates a Curie–Weiss behavior.

**Figure 8 materials-14-03692-f008:**
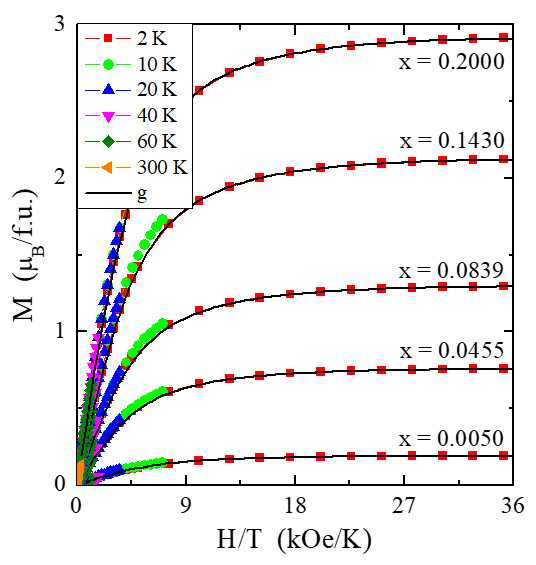
Magnetization *M* vs. *H*/*T* of CCGMWO solid solution for *x* = 0.0050, 0.0455 0.0839, 0.1430, 0.2000, and *y* = 0.02 recorded at 2, 10, 20, 40, 60, and 300 K. The solid (black) line indicates a Landé factor fit (g).

**Figure 9 materials-14-03692-f009:**
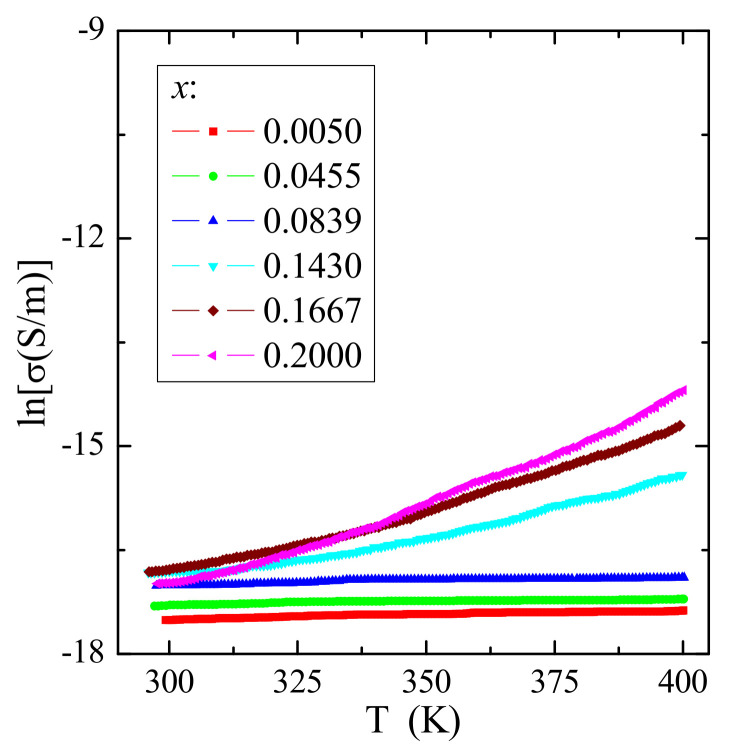
Electrical conductivity (lnσ) vs. reciprocal temperature 10^3^/*T* of CCGMWO solid solution.

**Figure 10 materials-14-03692-f010:**
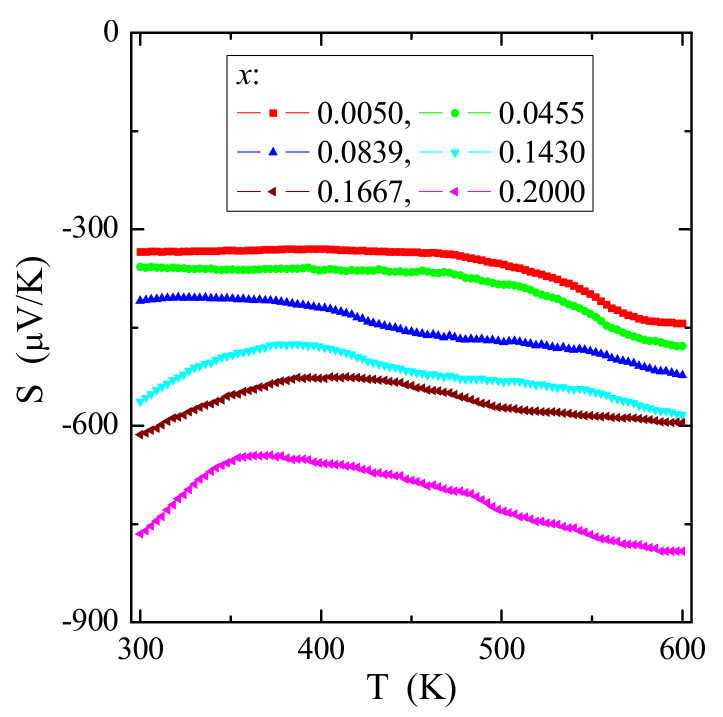
Thermoelectric power *S* vs. temperature *T* of CCGMWO solid solution.

**Figure 11 materials-14-03692-f011:**
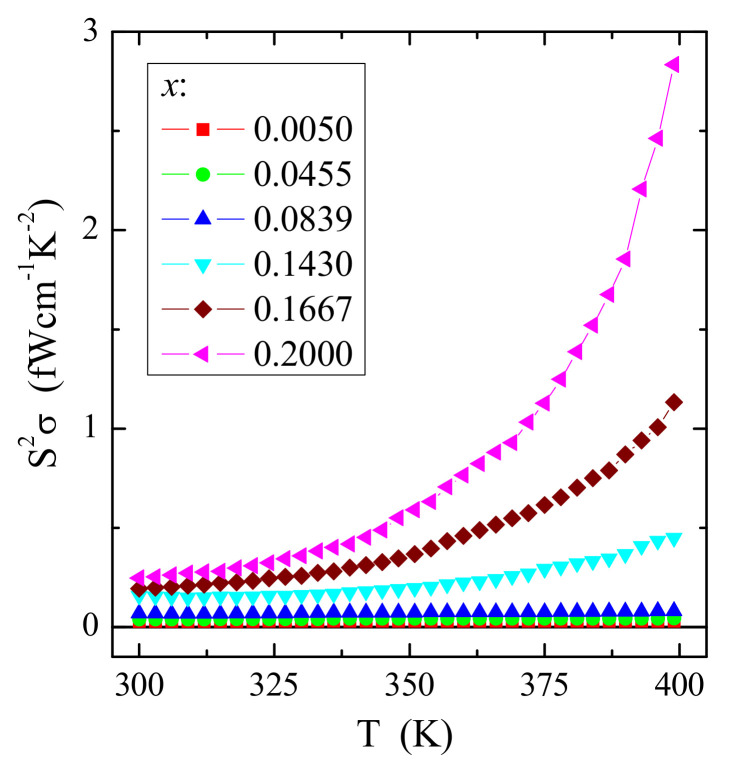
Power factor S^2^σ vs. temperature *T* of CCGMWO solid solution.

**Figure 12 materials-14-03692-f012:**
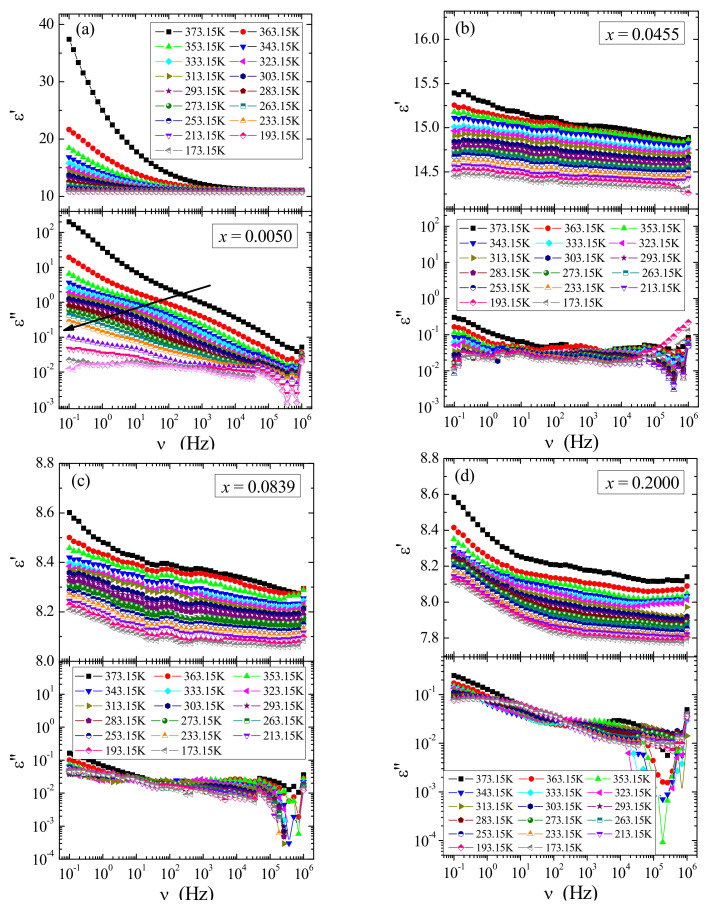
Real (ε′) and imaginary (ε″) part of permittivity vs. frequency *ν* of CCGMWO solid solution for *x* = 0.0050 (**a**), 0.0455 (**b**), 0.0839 (**c**), and 0.2000 (**d**) recorded in the temperature range of 173–373 K.

**Figure 13 materials-14-03692-f013:**
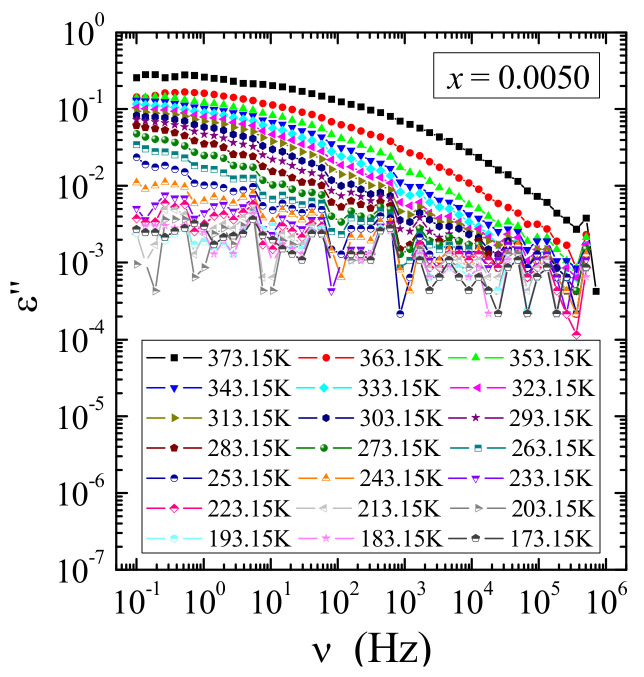
Calculated data from the Kramers–Kronig transformation of the real part of permittiity (ε′) vs. frequency ν of CCGMWO solid solution for x = 0.0050 to the ε″ representation recorded in the temperature range of 173–373 K.

**Figure 14 materials-14-03692-f014:**
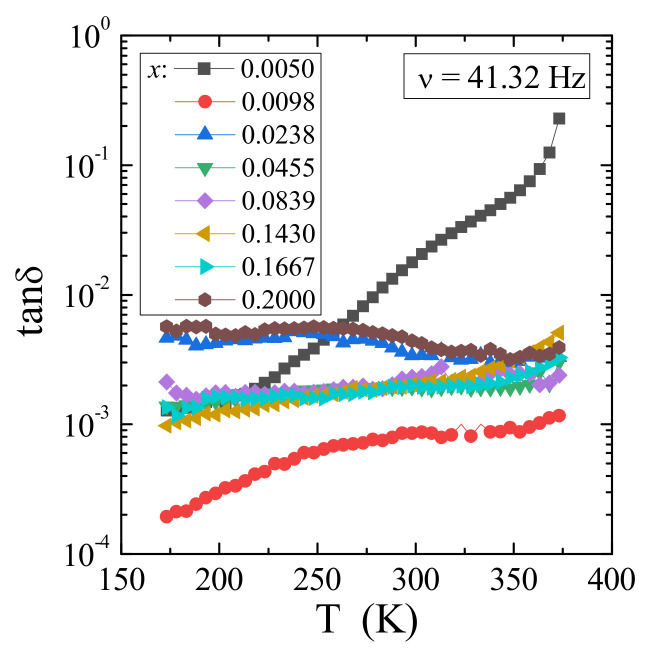
Loss tangent tan*δ* vs. temperature *T* of CCGMWO solid solution recorded at 41.32 Hz.

**Table 1 materials-14-03692-t001:** Lattice constants (*a*, *c*, and *c*/*a*), volume of unit cell (V), experimental density (d), determined optical direct band gap (E_g_), and Urbach energy (E_U_) of CCGMWO solid solution.

x	y	*a* (Å)	*c* (Å)	*c*/*a*	V (Å^3^)	d (g cm^−3^)	E_g_ (eV)	E_U_ (eV)
0	0	5.22836(12)	11.4379(4)	2.1877	312.663	4.25(1)	3.79	0.295
0.0050	0.02	5.22849(10)	11.4300(6)	2.1861	312.463	4.31(2)	3.78	0.319
0.0098	0.02	5.22797(14)	11.4340(7)	2.1871	312.509	4.36(1)	3.77	0.312
0.0283	0.02	5.22977(12)	11.4248(7)	2.1846	312.473	4.50(1)	3.73	0.340
0.0455	0.02	5.23360(9)	11.4244(4)	2.1829	312.922	4.69(2)	3.72	0.340
0.0839	0.02	5.23515(8)	11.4130(6)	2.1801	312.794	5.06(2)	3.63	0.413
0.1430	0.02	5.23718(10)	11.4162(5)	2.1798	313.124	5.65(2)	3.51	0.578
0.1667	0.02	5.23644(11)	11.4209(6)	2.1810	313.163	5.88(3)	3.57	0.422
0.2000	0.02	5.23606(10)	11.4457(4)	2.1859	313.799	6.17(1)	3.65	0.366

**Table 2 materials-14-03692-t002:** Magnetic parameters of CCGMWO solid solution.

x	y	C (emu·K/mol)	θ (K)	µ_eff_ (µ_B_/f.u.)	p_eff_	M_0_ (µ_B_/f.u.)	g
0.0050	0.02	0.197	0.4	1.254	1.229	0.192	1.40
0.0098	0.02	0.264	0.9	1.454	1.454	0.270	1.48
0.0238	0.02	0.489	0.3	1.978	1.969	0.462	1.60
0.0455	0.02	0.810	0.4	2.545	2.571	0.760	1.67
0.0839	0.02	1.880	1.1	3.332	3.384	1.300	1.70
0.1430	0.02	2.308	0.9	4.296	4.347	2.130	1.73
0.1667	0.02	2.671	0.5	4.622	4.678	2.460	1.76
0.2000	0.02	3.283	0.1	5.124	5.107	2.920	1.82

*C* is the Curie constant, θ is the Curie–Weiss temperature, *µ*_eff_ is the effective magnetic moment, *p*_eff_ is the effective number of Bohr magnetons, *M*_0_ is the magnetization at the highest value of *H/T*, and *g* is the Landé factor.

## Data Availability

Data sharing not applicable.

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
