# Peer review of "Effect of Gd3+ Substitution on Thermoelectric Power Factor of Paramagnetic Co2+-Doped Calcium Molybdato-Tungstates"

_materials, 2021, doi:10.3390/ma14133692_

Round 1
Reviewer 1 Report
The work is of scientific interest to specialists in the field of obtaining functional materials. The data are reliable and do not cause much doubt. Nevertheless, there are several points before the paper can be published. I hope that authors after revision can improve the paper and can published it in Materials. My decision is major revision. But I hope that after brutal revision it can be accepted. Really I impressed by the paper.
My comments:
- Please highlight in Introduction the motivation of the Gd concentration range (x and y) and step of concentration.
- It will be better adding formula of the chemical reaction for samples preparation from initial components.
- Authors compared ionic radius for Co and Gd ions in CN=8. What is the oxygen surrounding in objects? Only CN=8? May be it has octahedral coordination (CN=6)? And what are the ionic radii for other CN?
- I feel that it will be better place fig. and tab. in the text (not at the end). Using template of the MDPI.
- Fig. 6-13 are collapsed. Please make it accurate.
Reviewer 2 Report
Some figures of the manuscript are overlapped making it impossible to read the information presented in the figures.
I suggest the authors reorganized the figures and submit the manuscript again.
Reviewer 3 Report
The authors study a series of Co2+-doped and Gd3+-co-doped calcium molybdato-tungstates, basically changing the Gd mole fraction in the range below 0.2.
Standard techniques for synthesis and characterisation have been used and the theoretical background is minimal. The study does not lack techniques but I cannot say that I am impressed by its novelty.
The title propably refers to Figure 10, which shows that increase of the Gd mole fraction leads to an increase of the thermoelectric power factor. This is intersting, but it has been done for a constant mole fraction of Co, namely 0.02. What happens if we change this?
The two tables are useful.
The articles in not carefully written, with unacceptable figures, at least as they arrived to me. This ancient custom of putting the figure captions away from the figures should eclipse. It does not help the referee, which in turn, does not help the authors.
My opinion is that the article can be published after the authors have addressed these points.
Round 2
Reviewer 1 Report
Revised version can be accepted
Reviewer 2 Report
The authors have addressed the comments. This manuscript reported CCGMWO powders and characterized their optical, magnetic, electrical, and dielectric properties. The increment of thermoelectric power factor shows the potential applications in electronics. In my personal opinion, it is good for publication.